# Avian Oropharyngeal Trichomonosis: Treatment, Failures and Alternatives, a Systematic Review

**DOI:** 10.3390/microorganisms10112297

**Published:** 2022-11-19

**Authors:** María Teresa Gómez-Muñoz, Miguel Ángel Gómez-Molinero, Fernando González, Iris Azami-Conesa, María Bailén, Marina García Piqueras, Jose Sansano-Maestre

**Affiliations:** 1Department of Animal Health, Faculty of Veterinary Science, University Complutense of Madrid, 28040 Madrid, Spain; 2Grupo de Rehabilitación de la Fauna Autóctona y su Hábitat (GREFA), Monte del Pilar s/n, 28220 Majadahonda, Spain; 3Departmental Section of Pharmacology and Toxicology, Faculty of Veterinary Science, University Complutense of Madrid, 28040 Madrid, Spain; 4Department of Preventive Medicine and Public Health and Microbiology, Faculty of Medicine, Autonomous University of Madrid, 28029 Madrid, Spain; 5Escuela de Doctorado, Catholic University of Valencia, 46002 Valencia, Spain; 6Department of Animal Health and Public Health, Catholic University of Valencia, 46002 Valencia, Spain

**Keywords:** *Trichomonas gallinae*, treatment, resistance to treatment, avian trichomonosis, natural products, new drugs, delivery systems

## Abstract

Oropharyngeal avian trichomonosis is a potentially lethal parasitic disease that affects several avian orders. This review is focused on the disease treatments since prophylactic treatment is prohibited in most countries and resistant strains are circulating. A systematic review following the PRISMA procedure was conducted and included 60 articles. Successful and non-toxic treatments of avian oropharyngeal trichomonosis started with enheptin, a drug replaced by dimetridazole, metronidazole, ornidazole, carnidazole and ronidazole. Administration in drinking water was the most employed and recommended method, although hierarchy of the avian flocks and palatability of the medicated water can interfere with the treatments. Besides pigeons, treatments with nitroimidazoles were reported in budgerigars, canaries, finches, bald eagles, a cinereous vulture and several falcon species, but resistant strains were reported mainly in domestic pigeons and budgerigars. Novel treatments include new delivery systems proved with traditional drugs and some plant extracts and its main components. Ethanolic extracts from ginger, curry leaf tree and *Dennettia tripetala*, alkaloid extracts of *Peganum harmala* and essential oils of *Pelargonium roseum* and some Lamiaceae were highly active. Pure active compounds from the above extracts displayed good anti-trichomonal activity, although most studies lack a cytotoxicity or in vivo test.

## 1. Introduction

Oropharyngeal avian trichomonosis is a potentially lethal disease that can affect many bird species, although the most commonly affected are pigeons and raptors [1,2]. Yellow or white caseous lesions at the oropharynx are the commonest signs of the disease, although internal organs, ocular or nasal implications can be found [3]. Canaries (*Serinus canaria domestica*), budgerigars (*Melopsittacus undulatus*) and finches may appear vomiting, lethargic, or emaciated [4,5,6,7]. Commonly, young birds are more heavily affected by the disease [6,8,9].

Nitroimidazole drugs are the most effective and commonly used since the 1950s, when they were first tested against the parasite; however, since 1990, resistant strains appeared in frequently treated flocks or aviaries. Prophylactic treatments are prohibited in many countries and no vaccines are available. For that reason, new treatments are necessary. This review focuses on the most frequently employed treatment for the disease, the appearance of resistances to nitroimidazoles and the examination on data of potentially new treatments or delivery systems.

## 2. Materials and Methods

The present systematic review was conducted following the guidelines of the Preferred Reporting Items for Systematic Review and Meta-Analysis (PRISMA) [10]. The objective of the present review is to describe and analyze the treatments employed against oropharyngeal trichomonosis. Within this framework, specific objectives are: first, to review the employed treatments to prevent and treat avian oropharyngeal trichomonosis; second, to review the reported resistance to the standard treatments; and third, to review new options for treatments, including those from natural products, or new formulations of previously employed drugs.

### 2.1. Databases and Strategy of Search

For this review three databases were employed: PubMed (MEDLINE), Scopus and Web of Science. Three searches were performed to include the highest number of articles, with the following terms and Boolean connectors: (1) “*Trichomonas gallinae*” AND “Treatment”, (2) “*Trichomonas gallinae*” AND (“Metronidazole” OR “Nitroimidazole” OR “Carnidazole” OR “Dimetridazole” OR “Ronidazole” OR “Ornidazole” OR “Tinidazole”) AND (“Resistance” OR “Sensitivity”), and (3) “*Trichomonas gallinae*” AND (“Natural products” OR “Essential oils” OR “Natural compounds” OR “Plant products” OR “Plant extracts”). Articles with no restriction for the date of publishing were included. Additionally, a search within the cited references in each article was performed and related references included. The final search was carried out on 8 October 2022. Duplicates were removed from the list before adding references to Mendeley. The abstract of each article was manually reviewed and agreement about the inclusion of the articles in the systematic review was achieved by all the authors.

### 2.2. Reasons for Exclusion

The following criteria for exclusion were applied to the identified records: articles dealing with other trichomonads (not from the avian oropharynx) or other parasites, review articles, meeting abstracts, patents, potentially toxic products, studies using other techniques, articles that do not meet the appropriate scientific criteria (i.e., doses not specified, methods not sufficiently explained, results not addressed properly), books and book chapters. No automated tools for analysis were employed, since the number of retrieved articles was not very extensive.

## 3. Results and Discussion

### 3.1. Search Result

The search identified a total of 233 records, 43 from PubMed, 77 from Scopus and 113 from the Web of Science. After removal of duplicates, a total of 114 articles were manually examined, exclusion criteria applied, the references lists were manually checked and new records were included. A total of 60 articles were finally analyzed and discussed in the present review (Figure 1) (Appendix A).

Metadata statistical analysis was not carried out due to the diversity of methods employed by the different authors, but results from different studies are compared and conclusions obtained in each part of the manuscript.

### 3.2. Treatments against T. gallinae

Treatment of avian oropharyngeal trichomonosis has improved since the first attempts made during the 1940s and 1950s. The employment of copper sulfate, hydrochloric acid, Lugol’s solution or mercuric chloride were tested on naturally infected pigeons with anti-trichomonal (AT) effect, but prolonged doses, poor palatability, or adverse effects were noticed in most cases [11,12] and they were soon dismissed. Other substances with slight or null AT effect tested in vitro or in natural or experimental infections were: 1. furazolidone, only efficient at high doses with toxic effects [13], 2. penicillin and streptomycin [12], 3. auremycin, nitrophenide and sulfamides [14], 4. Osmaron B [8], and 5. nihydrazone [15].

Several drugs have been tested with success and low toxicity in the therapy of avian trichomonosis and most of them are still employed when clinical cases appear. The majority of the studies were performed by employing naturally or experimentally infected domestic pigeons (*Columba livia*) (Table 1, Appendix A). Nitrothiazoles were introduced in the decade of 1950 to treat trichomonads and other protozoal infections, starting with enheptin (also known as ANT or 2-amino-5-nitrothiazole) [14], and later with similar molecules or derivates. Enheptin was first tested under several conditions, including several doses, in naturally and experimentally infected pigeons, and with asymptomatic and symptomatic birds [14]. High efficacy was achieved with doses from 18 to 45 mg/kg/day administered for seven days, with lower doses being less effective and higher doses being toxic. The drug was also proved efficient at 0.125% (equivalent to 20 mg/bird/day) in aqueous solution for 6 days, both in natural and experimental infections, although some old pigeons appeared refractory to the treatment [8]. In an experimental procedure employing the highly pathogenic Jones´ Barn strain, a dose of 30 mg/bird/day for 14 days was necessary to reach 100% AT efficacy, and at this time, differences in drug sensitivity were observed between studies and apparently depending on the strains infecting the birds or the chronicity of the infection [16].

Metronidazole and dimetridazole were introduced in the 1960s and comparisons with enheptin were made, both in pill format and within drinking water, the latter being the recommended administration system by the authors [17]. Both drugs were effective at the dose of 50 mg/kg (approximately 15 mg/bird/day) for five days in chronically infected pigeons, or at 0.05% in drinking water administered for 3–5 days to naturally or experimentally infected pigeons [15,18], although temporary ataxia was observed in 12% of the birds if dimetridazole was administered for 6 days [15]. More trials were performed with dimetridazole, one of them administered between 5–10 mg/bird/twice daily for 3 days, with good results and no toxicity reported in experimentally infected pigeons [19], while another reported no efficacy if administered for 1 day (25 mg/day oral) but high and perdurable efficacy (up to 1 month) when administering the same dose for five days [20]. Finally, the combination of metronidazole (40–60 mg/day, 5–6 days) with oxytetracycline was effective in not severely affected pigeons [21]. It seems that the infection does not interfere with the pharmacokinetics of metronidazole [22]. In summary, five days is the recommended administration schedule to achieve 100% efficacy and perdurable results, with doses varying from 15 to 60 mg/bird/day.

Ornidazole was evaluated in the 1980s in infected pigeons, and under several formats of administration [23]. Doses of 2.5–5 mg/bird for 3 days dissolved in water were effective for at least seven days after treatment while lower doses were not. If administered in solid state, doses from 10 to 40 mg/bird/day for 3 days showed optimum AT effect for at least 17 days, the highest dose being the one recommended by the authors. Carnidazole is usually administered in one dose (10 mg/kg) and displayed lower efficacy when compared with metronidazole at the same dose in naturally infected pigeons (87.5% vs. 100%) [24]. When employed in a clinical situation, 10 mg/bird was enough to avoid clinical manifestations in adult pigeons; however, 60% of the squabs died of trichomonosis at 5 mg/bird [25]. The authors argued that early diagnosis was lacking in this case and was critical for treatment success. Nifursol 50% has been evaluated against experimental infections with *T. gallinae*. Doses of 600–1200 mg/kg of food were enough to achieve results comparable with the administration of metronidazole in drinking water [26].

**Table 1 microorganisms-10-02297-t001:** Treatment employed against avian oropharyngeal trichomonosis in domestic pigeons (*Columba livia*). Products, doses and success rates are indicated, as well as side effects. Only effective drugs (efficacy higher than 80%) are included in the table.

Product	Dose	Success Rate	Side Effects	Clinical Signs	Type of Infection	Reference
Enheptine	18–45 mg/kg/d, 7 d	100%	No	No	Natural	[14]
			No (83.3% mortality at 90–280 mg/kg/d)	Yes (controls died)	Experimental	
Enheptine	150 mg/bird	80–100% (lower efficacy in old birds)	No (at 500 mg/bird vomit and absence of mourning)	No	Natural	[8]
	0.125% DW, 6 d	86.7–100% (lower efficacy in old birds)	No	No	Most natural and some experimental	
Enheptine	10–30 mg/bird/d, 7 d (DW)	77.8–92.9%	No	Yes (controls died)	Experimental	[16]
	30 mg/bird/d, 14 d (DW)	100%	No	Yes (controls died)		
Enheptine	100 mg/bird	70–100%	No	No	Natural (in cages or flocks)	[17]
	0.1 % DW	100%	No	No		
Metronidazole	25 mg/bird	70–100%	No	No		
	0.1% in DW	100%	No	No		
Dimetridazole	24 mg/bird	80–100%	No	No		
	0.125% in DW * (product at 40%)	100%	No	No		
Metronidazole	50 mg/kg, 5 d	100%	No	No	Natural and Experimental	[15]
Dimetridazole	50 mg/kg, 5 d	100%	No	No		
	0.05% DW, 3–6 d	100%	Ataxia for 24 h at 6 days (11.7%)	No		
Dimetridazole	50 mg/kg, 5 d or 0.05% DW, 5 d	Effective	No	Weight loss, oropharyngeal lesions	Natural	[18]
Metronidazole	50 mg/kg, 5 d	Effective	No			
Ornidazole	2.5–5 mg/bird 3 d DW	Effective	No	No	Natural	[23]
	Solid: 10–40 * mg/kg, 3 d	Effective (lower doses not effective)	No	No		
Dimetridazole	200–400 * mg/L, 25 mL, 3 d	Effective at day 3	No	No	Experimental	[19]
Dimetridazole	25 mg/bird/d, 5 d	100% (not effective 1 dose)	No	No	Natural	[20]
Dimetridazole	10 mg/kg	100%	No	No	Natural	[24]
Carnidazole	10 mg/kg	87.5%				
Nifursol 50%	600–1200 mg/kg food	Effective	No	No	Natural	[26]
Carnidazole	10 mg/adult, 5 mg/squab	100% adults, 40% squabs	No	Apathy, anorexia, diarrhea, 60% mortality squabs	Natural	[25]
Metronidazole and oxytetracycline	40–60 mg/kg, 6 d	80%	No	Listless, oropharyngeal lesions, two deaths	Natural	[21]
Metronidazole	25 mg/kg, 5 d or 25 mg/kg iv	effective	No	No	Experimental	[22]

*: recommended treatment by the authors. d: days. DW: drinking water.

Besides domestic pigeons, other birds, wild or domestic, have been treated against naturally acquired oropharyngeal avian trichomonosis, with clinical manifestations, employing in most cases metronidazole or dimetridazole (Table 2). Several cases of wild birds are described in the literature. Debilitated bald eagles (*Haliaeetus leucocephalus*) with oropharyngeal lesions were successfully treated with dimetridazole at doses of 50 mg/kg or total doses of 375–500 mg/bird, and in all cases, 1 day of treatment was not enough to eliminate the infection, and the authors needed at least two or three doses separated by 1–8 days, depending on the situation, to eliminate the trichomonads [27,28]. Besides, one asymptomatic bald eagle was treated with 50 mg/kg/12 h of metronidazole for five days to eliminate the infection [29]. Additionally, a debilitated cinereous vulture (*Aegypius monachus*) with mild oral lesions was treated with metronidazole (50 mg/kg, five days), although the treatment needed to be repeated to eliminate the infection [30]. In these two last cases, *Trichomonas gypaetinii*, another oropharyngeal trichomonad, was the etiological agent. Metronidazole has been used to treat clinical cases of trichomonosis (lesions in orpharynx, sinuses and internal organs) in several species of falcons at 50 mg/kg/day for 5–7 days, although injectable dimetridazole (5%, 0.25 mL) was employed to treat infraorbital sinuses [3]. During an outbreak of oral trichomonosis in wood pigeons (*Columba palumbus*), the administration of dimetridazole at 200 ppm in the food of game bird feeders ended the mortality events [31].

Domestic animals other than pigeons have been treated with both drugs. Clinically affected budgerigars (*Melopsittacus undulatus*) have been treated with dimetridazole (0.05 mg/g bird for seven days) in drinking water [4] or metronidazole (30 mg/kg, ten days) [7]. Metronidazole was partially effective when administered at 200 mg/L of drinking water for five days to canaries with clinical signs of oral trichomonosis, and completely eliminated the infection at 200 mg/kg for five days [6]. Turkeys with clinical signs of trichomonosis and raised together with pigeons and hens also recovered from the disease with metronidazole in drinking water for seven to ten days [32].

Carnidazole has been less widely employed to treat avian oropharyngeal trichomonosis in birds other than pigeons (Table 2). House finches (*Haemorhous mexicanus*) were treated at 20 mg/kg/day for five days [5] with some debilitated birds dying of trichomonosis. Severely affected sparrowhawk chicks were treated employing carnidazole, but only 2/5 recovered from the infection [9]. Carnidazole was also tried at several doses in naturally infected pink pigeons (*Nesoenas mayeri*) [33]. The employment of 10 mg/bird in one dose or 5–10 mg/ bird every 2–4 days until clinical signs ceased seemed effective in short term, although the infection eventually returns. Ronidazole and dimetridazole were also tested in drinking water, but variable attendance at drinking station was observed by the authors, probably not reaching the appropriate dose [33]. It seems that the low palatability of some drugs, the hierarchy of pigeons, the problems to dissolve some drugs, or the time of the year could interfere with the efficacy of the treatments, and these facts should be considered when establishing a treatment protocol [8,33].

Severely affected young birds are more prone to die of trichomonosis even if treated at the appropriate dose, and this fact has been observed in squabs [25], but also in canaries [6] and raptor chicks [9]. Highly debilitated birds of other species, such as budgerigars [4], pigeons [21], or house finches [5] also died in spite of treatment administration. Prompt treatment seems to be crucial for the success of the therapy.

**Table 2 microorganisms-10-02297-t002:** Treatment employed against avian oropharyngeal trichomonosis in birds other than domestic pigeons. Products, doses and successful rates are indicated. Only effective drugs (efficacy higher than 80%) are included in the table.

Product	Dose	Successful	Species	Clinical Signs	Reference
Dimetridazole	47 mg/kg (2 d in 1 week)	Yes	Bald eagle (*Haliaeetus leucocephalus*)	Debilitated, oropharyngeal lesions	[27]
Dimetridazole	250–500 mg/d (3 d in 1 week)	Yes	Bald eagle (*Haliaeetus leucocephalus*)	Debilitated, oropharyngeal lesions	[28]
Metronidazole	50 mg/kg/12 h, 5 d	Yes	Bald eagle (*Haliaeetus leucocephalus*)	No	[29]
Metronidazole	50 mg/kg, 5 d (repeated)	Yes	Cinereous vulture (*Aegypius monachus*)	Mild lesion at oropharynx	[30]
Metronidazole	50 mg/kg 5–7 d (+dimetridazole 5% injectable in infraorbital sinuses)	Yes	Falcons (several species)	Oral, nasal lesions, some with organs and infraorbital sinuses affected	[3]
Carnidazole		Partially	Sparrowhawk (*Accipiter nisus*)	4/5 chicks with severe oropharyngeal lesions	[9]
Dimetridazole	0.05 mg/g bird DW, 7 d	Yes (emaciated birds died)	Budgerigars (*Melopsittacus undulatus*)	Vomit, emaciation	[4]
Metronidazole	30 mg/kg, 10 d	40% (40% dead)	Budgerigars (*Melopsittacus undulatus*)	Vomit, emaciation, oral lesions	[7]
Metronidazole	200 mg/L DW, 5 d or 20 mg/Kg/d, 5 d	effective	Canaries (*Serinus canaria*)	Lethargy, vomit, emaciation, death	[6]
Carnidazole	20–30 mg/kg, 5 d	Partially	House finches (*Haemorhous mexicanus*)	Emaciation, ocular and nasal discharge, emaciation	[5]
Dimetridazole	200 ppm in food 10 d	Effective	Wood pigeon (*Columba palumbus*)	Oropharyngeal lesions, organs affected	[31]
Carnidazole	5–10 mg/bird, various doses	Effective in short term	Pink pigeon (*Nesoenas mayeri*)	Wight loss, emaciation, oropharyngeal lesions	[33]
Ronidazole/Dimetridazole DW		Variable attendance			

d: days; h: hours. DW: drinking water.

### 3.3. Resistance of T. gallinae to Nitroimidazoles

Nitroimidazole drugs are the treatment of choice against avian oropharyngeal trichomonosis. However, failures in the success of treatments in birds infected by *T. gallinae* were first reported in 1990 [34]. Since then, nine studies had been conducted specifically to demonstrate the existence of resistant strains to these compounds (Appendix A). Experiments had been carried out using metronidazole, dimetridazole, ronidazole, carnidazole, ornidazole and tinidazole. These studies showed that there are resistant strains of *T. gallinae* to all the nitroimidazole drugs used in birds [22,34,35,36,37,38,39,40,41]. However, there are not studies directed to elucidate the mechanisms by which *T. gallinae* acquires resistance to nitroimidazole drugs. It has been assumed that the molecular mechanisms of resistance could be similar to *T. vaginalis*, since they are closely related. Some authors suggested that direct contact between the ingested drug and trophozoites within the lumen of the oral cavity or crop is required for complete protozoocidal activity [36]. Nitroimidazoles are frequently administered in drinking water, passing directly to the proventriculus in absence of crop content. The administration of these drugs with food would enhance the contact of the drug with the parasite, since food reduces the time to empty the crop, improving the response to the treatment [36].

#### 3.3.1. In Vivo Resistance

The first report on *T. gallinae* resistance to nitroimidazoles appeared after noticing treatment failures in clinically ill infected birds. The authors unsuccessfully tried four consecutive treatments in five pigeons from a flock refractory to previous treatments. The sequential administration of dimetridazole, ronidazole, carnidazole and metronidazole did not eliminate the parasite from the animals, and the authors suspected acquired resistance [34].

Between 2.5 and 5 times the recommended dosage (40 mg/L drinking water) of ronidazole were evaluated in pigeons carrying drug-resistant *T. gallinae* strains. The resistance of the strains was previously tested in in vitro experiences. Only the highest dosages (200 mg/L) eliminated the infection in the animals [35].

Carnidazole and dimetridazole resistance were assessed in a flock frequently treated by the owner with these drugs [38]. The efficacy of the treatment was very low with both molecules, since only 19% of the animals treated with carnidazole and none of the treated with dimetridazole eliminated the infection.

#### 3.3.2. In Vitro Resistance

Several studies demonstrated the existence of resistant strains of *T. gallinae* obtained from different hosts, under laboratory conditions. The parasites were exposed to different concentrations of drugs in order to obtain the minimum inhibitory concentration (MIC), also named as minimum lethal concentration (MLC) or the French expression “Concentration Minimale Trichomonacide” (CMT) [37], but results vary according to the methodology employed (parasite growth and inverted microscopy), although the absence of motility observed by inverted microscopy is the most advisable technique [38] (Table 3, Appendix A).

In a different experiment, isolates obtained from racing pigeons were employed to evaluate the effectiveness of metronidazole, dimetridazole, ronidazole and carnidazole, and the authors encountered resistance to one or more drugs in six out of eight isolates [35]. The pattern of resistance was variable among isolates but no resistance to dimetridazole was reported in any of the isolates investigated. In addition, ronidazole was selected as the drug more suitable for treatments in vivo since it showed the lowest levels of resistance. Ronidazole was also evaluated for drug resistance in 31 isolates of *T. gallinae* from pigeons [37]. A MIC value of 80 µg/mL was established as resistance threshold for the authors. However, they observed that 45% of the isolates had a MIC value of 40 µg/mL, a little below the threshold was considered as an indicator of resistance in vivo. The authors found a relationship between the frequency of treatments and the decrease in the susceptibility of the strains to ronidazole.

Ronidazole was again the drug with higher efficacy against the parasite in the experiments conducted by Munoz et al. [38]. After studying resistance of four isolates of *T. gallinae* recovered from different birds (wild and domestic) against metronidazole, dimetridazole, ronidazole, carnidazole and ornidazole determined that the MLC varied between 1.7–7.1 µg/mL, except for an isolate from a racing pigeon which required concentrations at least eight times higher for all the drugs used.

McKeon et al. (1997) [36] studied the MIC of ronidazole, metronidazole and dimetridazole in six isolates obtained from budgerigars and a Senegal dove (*Spilopelia senegalensis*). In their experiments, the authors tried to emulate the time that the parasite is in contact with the drug in the crop, by limiting the time that parasites were in medicated medium. They obtained MIC values similar for the three drugs, ranged from 40 to 70.5 µg/mL for metronidazole, 30 to 50 µg/mL for dimetridazole and 40 to 60 µg/mL for ronidazole. However, the MIC of the isolate obtained from the dove was higher than the MIC of the isolates obtained from budgerigars, being 96, 80 and 92 µg/mL for metronidazole, dimetridazole and ronidazole, respectively.

Variations in resistance patterns has been observed not only between strains but also between clones obtained from the same sample [39]. The authors obtained two strains genetically different from the same bird (a racing pigeon), which displayed significantly different drug sensitivities. In addition, genetic relationships between resistance phenotype and ITS/5.8S/ITS sequence was detected. This association was observed by Rouffaer et al. [41] comparing isolates obtained from wild and racing pigeons. In both studies, resistant strains were more frequently associated with genotype A of this sequence (equivalent to genotype C of other authors [42]), which has been found at higher prevalence in pigeons [43].

Results observed by different authors show relationships between nitroimidazole resistant strains, hosts and genetic diversity of *T. gallinae*. Pigeons, mainly racing pigeons, seem to be an important source of resistance strains [38,39,41]. The reasons can be diverse, but continuous preventive treatments with underdosed treatments, as occurs frequently in drink water dosages, could partially explain these questions.

### 3.4. Alternatives to the Traditional Treatment of Oropharyngeal Avian Trichomonosis

#### 3.4.1. Old Drugs, New Molecules and Novel Delivery Systems

There have been limited efforts in the search of new drugs to fight *T. gallinae*, since only six articles were found exploring the anti-trichomonal (AT) activity with novel drugs or novel delivery systems (Table 4, Appendix A). Only one of the studies employed a new molecule (chitosan) against *T. gallinae* [44], while four explored new carriers to load them with metronidazole [45,46,47,48], and another one loaded the carrier with rhodanine [49].

All the articles performed in vitro experiments and metronidazole was used as a reference compound. Most authors determined the AT activity within 1–6 h, while only one article evaluated the AT activity at 12, 24 and 48 h post addition of the compound [48]. Most of the articles employed the percent of growth inhibition by counting trophozoites, while one article employed the percent of mortality [44] and another employed the IC_50_ (50% lethal dose, dose) value to evaluate the AT activity [48]. None of them assessed cytotoxicity, except Tabari et al. [48], who did not find significant cytotoxic activity. The authors also employed in vivo experiments, proving the absence of hepatotoxicity, as well as high efficacy in eliminating trophozoites in experimentally infected pigeons at days 2 and 3 of treatment with metronidazole and nanolactoferrin-metronidazole at 50 mg/Kg for 5 days.

Chitosan is the only molecule evaluated as a new anti-*T. gallinae* compound. It is a molecule derived from chitin, a natural carbohydrate polymer mainly found in the skeleton of arthropods and cell walls of fungi, and has been suggested as antimicrobial, antifungal and antioxidant [44]. The authors employed concentrations ranging from 125 to 1250 µg/mL and found percentages of mortality higher than 80% after 1 h of exposure and from 96.3% to 100% after 6 h. This molecule was later employed, together with other compounds (cellulose nanofibrils and tannic acid), as part of a pharmacological vehicle (named nanocarriers) loaded with metronidazole [45]. They found the carrier also active against *T. gallinae*, but with 30–50% lower AT values. The authors suggest a possible cytotoxic effect of the carrier, although no experiments were conducted to evaluate this point.

Other drug delivery systems employed to deliver AT drugs were whisker-like SBA-15 nanocrystals (nanowhiskers) [46], Zeolite Y nanoparticles with tannic acid [47], zinc oxide [49] and nanolactoferrin [48]. Zinc oxide nanoparticles displayed higher AT activity than the coated nano particles with the drug [49]. None of the remaining delivery systems, except lactoferrin which was not tested alone, displayed significant AT activity. When evaluated at early times of the experiments, all of them reached 100% of growth inhibition at 3 h, although poly(rhodanine) coated with zinc oxide was faster in reaching this percentage of growth inhibition; however, this might be due to the zinc oxide, as we have mentioned before [49] (Table 4). However, higher doses and a different compound were used in this study and no direct comparison can be made with the rest.

All the experiments employing novel drug delivery systems showed good and fast AT activity (high mortality rate, high GI%) with the vehicles and concentrations employed when compared with metronidazole control (50 µg/mL), although the concentrations used for the carriers (mostly 2, 1 and 0.5 mg/mL) were, in general, much higher than the employed for metronidazole (Table 4), with the exception of nanolactoferrin, in which the same dose was applied for the delivery system and for metronidazole, and similar results were obtained with the drug alone or in combination with the carrier by the authors [48].

#### 3.4.2. Natural Products from Plants

An increasing interest is arising in natural products, among them those isolated from plants, since they are associated with the amelioration of life expectancy and life quality due to their composition, especially antioxidant molecules [50]. Besides this effect, many of them have been proven as anti-cancer, antibacterial, antifungal and anti-parasitic, including protozoa, helminths and arthropods [40]. In this section, we expose all the studies testing natural products and derivates as anti-*T. gallinae*.

Seventeen articles were found after the systematic review dealing with plant extracts or derived products. Of them, six employed methanolic extracts [50,51,52,53,54,55], five tested essential oils (EOs) [56,57,58,59,60], two are based on ethanolic extract [61,62], one tested alkaloid extracts [63], one tested an aqueous extract [40], and another one tested a dichloromethane extract [50] against *T. gallinae* (Table 5). Two of the studies evaluated only pure compounds or derivates [64,65], while three of the articles testing extracts also employed pure compounds as anti-*T. gallinae* [50,52,60] (Table 5, Appendix A).

Comparison among the in vitro studies is not an easy task, because of the lack of uniformity in the employed methods. To test the activity or vitality of the trophozoites, the most employed technique is counting in triplicate, while only one article measured the metabolic activity by employing the MTT method adapted to the protozoa [60]. Many of them employed the Inhibitory Concentration 50 (IC_50_, the dose needed to kill 50% of the trophozoites) to evaluate the activity against the protozoa, while a few used the Minimum Inhibitory Concentration (MIC). Most articles determined the in vitro anti-trichomonal activity at 24 h, but some of them tested the products at 24, 48 and 72 h. When examining the anti-*T. gallinae* effects at different times, the product/compound evaluated is described as trichomonastatic when IC_50_ values are higher at later times (48 h) and trichomonacidal if IC_50_ values are higher at earlier times (24 h) [53]. Since 24 h is used by most of the authors, we have employed this reference in Table 5 to show comparable measures.

All the studies included in vitro assays employing different doses of the extracts, while eight of them tested the extracts also in vivo [40,50,51,57,58,59,63,64]. Among the in vivo studies, three investigated the toxicity in rats or mice and found no notable toxic effects [50,51,64], while the others compared the effect of the extract with the administration of metronidazole in experimentally infected birds (pigeons mostly). Scarce or no toxicity of the extracts and comparable effectivity to metronidazole was obtained [40,57,58,59,63].

The method of extraction most frequently employed was methanol extraction, followed by essential oils extraction, while the other methods were employed only in one or two articles. The most studied plant as anti-*T. gallinae* was the curry leaf tree (*Murraya koenigii*), probably due to several experiments made by the same authors [51,52,64]. Other plants were also employed in more than one article, such as lavender (*Lavandula angustifolia*) [60,61], rosemary (*Rosmarinus officinalis*) [54,60] and garlic (*Allium sativum*) [40,54].

To compare the strength of the anti-trichomonal activity of the extracts, limits has been established in Table 5 as follows: for MIC and IC_50_: IC_50_ ≤ 50 µg/mL (High), IC_50_ > 50 and ≤400 µg/mL (Moderate), IC_50_ > 400 µg/mL (Low). According to the IC_50_ dose reported by the authors, the most effective extracts were the ethanolic extracts obtained from ginger (*Zingiber officinale*) and lavender (*Lavandula angustifolia*) [61], the essential oil from the leaves of *Dennettia tripetala* [56], the methanolic extracts of the seeds and pericarp from the curry leaf tree (*M. koenigii*), the alkaloid extracts from the seeds of *Peganum harmala* [63], the essential oil from *Artemisia sieberi* [57], and the essential oil of *Pelargonium roseum* [59]. Satisfactory results were also obtained from methanolic extracts of leaves and stem of *M. koenigii* [51], methanolic extract from the leaves of *Eugenia uniflora* [53], and essential oils from *Lavandula luisieri* and other Lamiaceae [60]. The rest of the extracts needed high doses to produce the desired anti-trichomonal effect (Table 5).

Cytotoxicity was not examined in many of the extracts employed, although in some cases it was examined on the pure compounds isolated from the tested extracts [60]. When examined, cytotoxicity was low in most of the assays, although some of them displayed moderate toxicity and caution should be taken in these cases [51] and a selectivity index was applied. This index measures the relative anti-trichomonal activity in comparison with the cytotoxic activity [51,60].

Pure compounds have been evaluated against *T. gallinae* in five articles and results are shown in Table 6. None of the compounds were tested against *T. gallinae* in experimental infections in vivo, although imperatorin, phellopterin, 3-phormylcarbazole and chalepin were proven to be safe when tested as anti-diabetic compounds [50], and others, such as carvacrol and thymol, have been used in animals for other reasons (antimicrobial, antioxidant, antiparasitic, and others, not shown in this review). Considering the anti-trichomonal strength, compounds present in the methanolic and dichloromethane extracts from the curry leaf tree (*M. koenigii*) and the fools curry leaf tree (*C. lansium*) are good alternatives in comparison with metronidazole, although the nitroimidazole offers lower IC_50_ values in most studies [50,52,64]. Carbazole alkaloids present in *M. koenigii*, such as girinimbirol, girinimbine, mahanimbine, mahanine, koenimbine, mahanimbirol, murrayanine, showed IC_50_ lower than 5 µg/mL, and furocoumarins from the same plant, such as isogosferol, indicolatone, isoimperatonin, bergapten, β-sitosterol and imperatorin displayed IC_50_ values lower to 10 µg/mL. These compounds were also evaluated in cytotoxicity assays, although the detailed IC_50_ dose for cytotoxicity was not shown. Similar compounds from *C. lansium*, such as imperatorin, phellopterin, 3-phormylcarbazole and chalepin have been tested in vitro for their effects as anti-diabetics, and no toxic effects were observed [50]. Harmine is an alkaloid present in the alkaloid extract of *C. lansium*, an extract highly efficient in eliminating the disease in birds. The alkaloid showed anti-trichomonal results similar to metronidazole in vitro (MIC = 30 µg/mL vs. MIC = 50 µg/mL of metronidazole), although the authors suggest that the employed isolate was resistant to the conventional treatment [63]. Finally, a series of compounds present in several Lamiaceae plants have been tested against *T. gallinae*. Considering the anti-trichomonal strength and the cytotoxicity, the most active against the parasite will be carvacrol (pure compound or nanoemulsion) [65], linalyl acetate, thymol, 4-terpinenol, γ-terpinene, p-cymene and D-fenchone [60]. Other compounds highly active against *T. gallinae*, such as camphene, β-caryophyllene, α and β-thujones, and α and β-pinenes, displayed moderate cytotoxicity and this fact should be further studied before recommending their use [60].

The strains employed for the assays can influence the obtained results. Most authors obtain MIC or IC_50_ values for metronidazole lower than 2 µg/mL at 24 h [50,51,52,60,62,64,65], while others employed strains with MIC or IC_50_ values higher than 10 µg/mL [53,59], or even higher than 15 µg/mL [40,57,61,63] which are considered resistant strains [41].

## 4. Conclusions

Several drugs have been evaluated against avian oropharyngeal trichomonosis with success, such as enheptin, dimetridazole, metronidazole, carnidazole, ronidazole and ornidazole.

Dimetridazole and metronidazole are the most employed drugs, probably due to their affordable price. Five days is the most recommended schedule administration, and administration in solid state, or mixed with food or drinking water have been assessed. Some authors recommended that mixing the drugs with food is a better choice, since the crop empties slower, and the flavor of drinking water can be altered and birds refuse to drink.

Since several strains resistant to nitroimidazoles are circulating, new efforts to search for alternative treatments or delivery systems are necessary. New treatments are mainly based on natural products, including chitosan or several extracts or essential oils obtained from plants, and their major components. Extracts and pure compounds from *Murraya koenigii*, *Dennettia tripetala*, *Clausena lansium*, *Peganum harmala*, *Eugenia uniflora*, *Artemisia sieberi*, *Pelargonium roseum*, *Zingiber officinale*, *Lavandula angustifolia* and other Lamiaceae are the most active against *T. gallinae*.

More cytotoxicity tests and in vivo studies are necessary to further elucidate the utility of the new proposals as alternative treatments for avian trichomonosis.

## Figures and Tables

**Figure 1 microorganisms-10-02297-f001:**
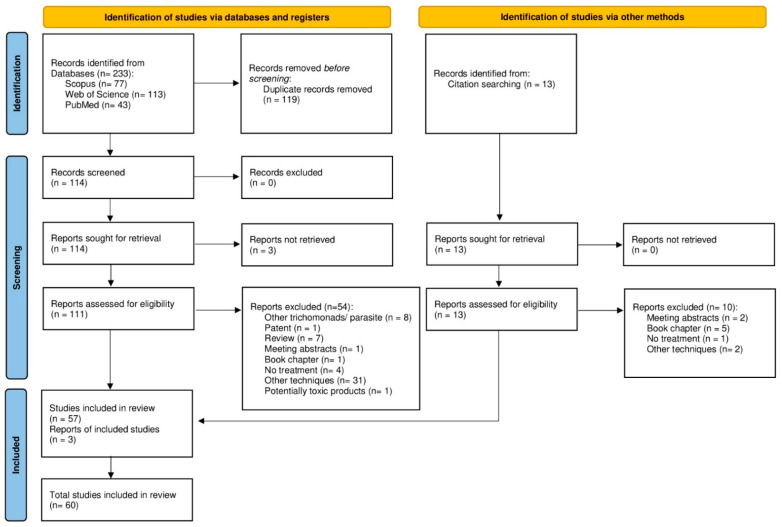
PRISMA flowchart employed for the systematic review of avian oropharyngeal trichomonosis treatments, failures and alternatives. The search of records was performed in PubMed, Scopus and Web of Science employing the terms and criteria conditions previously described.

**Table 3 microorganisms-10-02297-t003:** Minimum Inhibitory Concentration (MIC) in μg/mL of different nitroimidazole drugs evaluated against trichomonads isolated from avian oropharynx. The name of the strains and the host of which isolates were obtained are included.

Strain	Host	MIC (μg/mL)						Ref.
		Dimetridazole	Ronidazole	Carnidazole	Ornidazole	Tinidazole	Metronidazole	
Not named (*n* = 6)	Budgerigar	30–50	40–60	--	--	--	40–50	[36]
HF26	Bonelli’s eagle	7.8	1.9	7.8	5.8	--	7.8	[38]
79P	Rural pigeon	7.8	1.4	3.9	2.9	--	5.8	
P2	Urban Pigeon	3.9	1.9	3.9	3.9	--	7.8	
1FG	Racing pigeon	187.5	31.2	93.7	125	--	500	
Not named (*n* = 31)	Pigeon	--	80	--	--	--		[37]
5895-C1/06	Budgerigar	2.5 ± 0.3	6.7 ± 1.7	--	6.7 ± 1.7	--	2 ± 0.3	[39]
15935-C3/06	Budgerigar	3.0 ± 0.7	2.7 ± 0.3	--	2.0 ± 0.6	--	2.7 ± 0.3	
231-C1/07	Racing pigeon	5.0 ± 0.0	5.0 ± 0.0	--	16.7 ± 1.7	--	8.8 ± 1.3	
Austria/ 231-C3/07	Racing pigeon	33.3 ± 3.3	65.0 ± 2.9	--	76.7 ± 3.3	--	28.3 ± 1.7	
7895-C2/06	Racing pigeon	36.7 ± 3.3	61.7 ± 1.7	--	73.3 ± 6.7	--	25.0 ± 2.9	
8855-C6/06	Racing pigeon	83.3 ± 6.7	83.3 ± 6.7	--	103.3 ± 3.3	--	103.3 ± 3.3	
Not named	Pigeon	--	--	--	--	--	50 (24 h); 25 (48 h); 12.5 (72 h)	[40]
Not named (*n* = 15)	Wild pigeon	0.98–125	0.98–500	1.95–250	--	0.98–62.5	0.98–125	[41] *
Not named (*n* = 16)	Racing pigeon	1.95500	1.95–500	3.9–500	--	1.95–250	1.95 >500	
Not named (*n* = 5)	Pigeon	--	--	--	--	--	50 (12 h); 25 (24 h); 6.25 (36 h); 1.5 (48 h)	[22]

* Authors determined a threshold of 15.6 μg/mL to discriminate resistance for all the drugs evaluated. --: Not tested. Ref.: reference.

**Table 4 microorganisms-10-02297-t004:** Novel treatments against *T. gallinae*, including novel formulations evaluated employing in vitro experiments. %GI: percentage of growth inhibition. IC_50_ (µg/mL): concentration needed to produce 50% trophozoite mortality. MIC (µg/mL): minimum inhibitory concentration: concentration at which no movement of trophozoites was observed.

			% GI Carrier + Compound/% GI Carrier				
Compound	Carrier (Loading-Entrapment Efficiency)	Concentration	0–1 h	1–2 h	2–3 h	6 h	Reference
Chitosan	-	125 µg/mL	>80/-	-/-	>80/-	96.3/-	[44]
		250 µg/mL	>80/-	-/-	>80/-	>96.3/-	
		500 µg/mL	>80/-	-/-	>80/-	>96.3/-	
		1250 µg/mL	>80/-	-/-	>80/-	100/-	
MTZ	Chitosan nanocapsule-cellulose nanofibrils-tannic acid composites (64.56%) †	2 mg/mL	30–100/0–55	100/55–75	100/75–85	-/-	[45]
		1 mg/mL	0–50/0–25	50–100/25–45	100/45–75	-/-	
		0.5 mg/mL	0–30/0	30–60/0–25	60–100/25–60	-/-	
MTZ	-	50 µg/mL	0	0–15	15–37	-/-	
MTZ	Nanowhiskers	2 mg/mL	0–5/0	5–90/0	90–100/0–5	-/-	[46]
		1 mg/mL	0/0	0–60/0	60–100/0	-/-	
		0.5 mg/mL	0/0	0–50/0	50–100/0	-/-	
MTZ	-	50 µg/mL	0	0–15	15–37	-/-	
MTZ	Zeolite Y nanoparticles with tannic acid (69.9%)	2 mg/mL	0/0	0–72/0	72–100/0	-/-	[47]
		1 mg/mL	0/0	0–25/0	25–100/0	-/-	
		0.5 mg/mL	0/0	0–15/0	15–85/0	-/-	
MTZ	-	50 µg/mL	0	0	0–37 /7	-/-	
Poly(rhodanine)	Zinc oxide	10 mg/mL	0–65/0–100	65–100/100	100/100	-/-	[49]
		5 mg/mL	0–40/0–85	40–70/85–100	70–100/100	-/-	
		2.5 mg/mL	0–30/0–75	30–50/75–100	50–100/100	-/-	
MTZ	-	50 µg/mL	0	0–15	15–37	-/-	
			IC_50_/MIC 12 h	IC_50_/MIC 24 h	IC_50_/MIC 48 h	Cytotoxicity (fibroblast)	
MTZ	Nanolactoferrin (55%)		-/100	0.995/12.5	-/1.5	No	[48] *
MTZ	-		-/50	0.936/25	-/1.5	No	

* Also in vivo experiments, with no hepatotoxic effects. MTZ: metronidazole. †: possibly cytotoxic. -: not tested.

**Table 5 microorganisms-10-02297-t005:** Plant extracts employed as anti-*T. gallinae* in vivo.

Plant (Extract, Part of The Plant)	IC_50_/MIC 24 h	IC_50_/MIC 24 h MTZ	ATS	Cytotoxicity (IC_50_)/Test	Major Components (>5%, Ordered from Higher to Lower)	Ref.
*Murraya koenigii* (ME, leaf)	34/-	0–1/ -	H	M: 61.5	Yes (mahanimbine, girinimbine, isomahanimbine, murrayazoline, murrayazolidine, mahanine, % not shown)	[51]
*Murraya koenigii* (ME, stem)	25/-	0–1/ -	H	H: 14.5 /Brine shrimp lethality test	Yes (mahanimbine, mahanimbilol, girinimbine, murrayanine, murrayacine, murrayaquinone-A, % not shown)	
*Murraya koenigii* (ME, seed)	1.9/-	1.9/-	H	L: 750	No	[52]
*Murraya koenigii* (ME, pericarp)	2/-	1.9/-	H	L: 1500 /Hemagglutination Test		
*Dennettia tripetala* (EO, leaf)	0.13% *v/v*/-	1.02% *v/v*/-	H	N.t.	Yes (2-phenylnitroethane, linalool)	[56]
*Harungana madagascariensis* (EE, stem bark) *	187/-	1.87/-	M	N.t.	No	[62]
*Clausena lansium* (DME) *	19/-	1.9/-	H	N.t.	No	[50]
*Clausena lansium* (ME, leaf) *	2/-	1.9/-	H			
*Eugenia uniflora* (ME, leaf)	61.7/-	13.7/-	M	L: 260	No	[53]
*Eugenia uniflora* (several unidentified subfractions)	4.8–70.5/-	13.7/-	H-M	L. 230–3000 /Hemagglutination		
*Allium sativum* (WE) *	75,000	-/50	L	N.t.	No	[40]
*Quercus persica* (ME)	2500 (100% mortality)	1250 (100% mortality)	L	N.t.	No	[54]
*Allium sativum* (ME)	No effect		L			
*Artemisia annua* (ME)	5000		L			
*Myrtus communis* (ME)	5000		L			
*Rosmarinus officinalis* (ME)	5000		L			
*Zataria multiflora* (ME)	5000		L			
*Peganum harmala* (AE, seed) *	-/15	-/50	H	N.t.	No	[63]
*Artemisia sieberi* (EO) *	-/10	-/20	H	N.t.	Yes (α-thujone, β-thujone, camphor, 1,8-cineole, camphene)	[57]
*Pelargonium roseum* (EO) *	-/20	-/10	H	N.t.	Yes (β-citronellol, geraniol, linalool)	[59]
*Pulycaria disenterica* (ME, aerial)	6250 (100% mortality 6 h)	100 (100% mortality 6 h)	L	N.t.	No	[55]
*Lycopus europaeus* (ME, aerial)	28,370 (100% mortality 6 h)		L	N.t.	Np	
*Zingiber officinale* (EE)	-/25	-/50	H	N.t.	Yes (α-curumene, α-gigeberene, gingerol, ciclohexane, α-phernesene, cis-6-shagole)	[61]
*Lavandula angustifolia* (EE)	-/50		H	N.t.	Yes (linalool, borneol, α-pinene, 1,8-cineol, ocimene, linalyl acetate)	
*Cymbopogon flexuosus* (EO) *	220/528	N.t. Only one dose (880 µg/mL)	L	N.t.	Yes (β-geranial, z-citral, geraniol)	[58]
*Cymbopogon flexuosus* (nanoemulsion EO) *	110/418		M	N.t.		
only the best result for each plant is shown				N.t.		[60]
*Santolina chamaecyparissus* (EO, aerial)	394.3/-	1	L		No	
*Dittrichia graveolens* (EO, aerial)	259.7/-		L		No	
*Lavandula lanata* (EO, aerial)	731.7/-		L		No	
*Lavandula luisieri* 1 (EO, aerial)	189.8/-		M		Yes (camphor, trans-α-necrodyl acetate, lavandulyl acetate)	
*Lavandula luisieri* 2 (EO, aerial)	103.4/-		M		Yes (trans-α-necrodyl acetate, lavandulol, germacrene D, unidentified)	
*Lavandula angustifolia* (EO, aerial)	600.4/-		L		No	
*Lavandula x intermedia “Abrial”* (EO, aerial)	406.8/-		L		No	
*Lavandula x intermedia “Super”*(EO, aerial)	321.2/-		L		No	
*Lavandula angustiflia var. maillette* (EO, aerial)	373.9/-		L		No	
*Origanum virens* (EO, aerial)	175.4/-		M		Yes (γ-terpinene, linalool, linalyl acetate)	
*Origanum majorana* SD (EO, aerial)	139.8/-		M		Yes (carvacrol, p-cymene, β-caryophyllene)	
*Origanum majorana* HD (EO, aerial)	158.8/-		M		Yes (4-terpineol, γ-terpinene, α-terpinene, sabinene, p-cymene, ocimene, β-caryophyllene)	
*Rosmarinus officinalis* (EO, aerial)	256.6/-		L		No	
*Satureja montana* SD (EO, aerial)	141.4/-		M		Yes (carvacrol, γ-terpinene, p-cymene, thymol)	
*Mentha suaveolens* (EO, aerial)	303/-		L		No	
*Salvia officinalis* SD (EO, aerial)	139.1/-		M		Yes (β-thujone, 1,8-cineole, β-pinene, β-caryophyllene, viridiflorol, α-humulene)	
*Salvia hibrid* (*S. officinalis* x *S. lavandulifolia*) SD (EO, aerial)	134.6/-		M		Yes (β-pinene, 1,8-cineole, β-caryophyllene, camphene, trans-bornyl acetate, camphor, α-pinene)	
*Salvia sclarea* HD (EO, aerial)	117.4/-		M		Yes (linalyl acetate, linalool, α-terpineol, neryl acetate, germacrene D)	
*Thymus vulgaris* HD (EO, aerial)	166/-		M		Yes (thymol, p-cymene, γ-terpinene)	
*Thymus vulgaris* SD (EO, aerial)	193.3/-		M		Yes (p-cymene, thymol, β-caryophyllene)	
*Thymus zygis* HD (EO, aerial)	133.1/-		M		Yes (thymol, p-cymene, γ-terpinene, borneol, linalool, carvacrol)	

* Extracts evaluated also in vivo. MTZ: Metronidazole as a reference compound. IC_50_: Inhibitory Concentration 50. MIC: Minimum Inhibitory Concentration. ME: methanolic extract. EE: ethanolic extract. EO: essential oil. AE: alkaloid extract. WE: water aqueous extract. DME: dichloromethane extract. N.t.: not tested. SD: EO extracted by steam distillation. HD: EO extracted by hydrodistillation. Ref: reference. ATS: Anti-trichomonal strength is expressed according to MIC or IC_50_ values (µg/mL). H: high, IC_50_ or MIC ≤ 100 µg/mL, M: moderate, IC_50_ > 100 and ≤400 µg/mL, L: low IC_50_ > 400 µg/mL. For cytotoxicity (µg/mL) L: low IC_50_ > 100 µg/mL, M: moderate IC_50_ > 50 and ≤100 µg/mL, or H: high IC_50_ ≤ 50 µg/mL.

**Table 6 microorganisms-10-02297-t006:** Pure compounds employed as anti-trichomonal in vivo. ATS: Anti-trichomonal strength is expressed according to MIC or IC_50_ values.

Plant	Compound	IC_50_ 24 h/MIC 24 h	IC_50_ 24 h/MIC 24 h MTZ	ATS	Cytotoxicity (IC_50_ µg/mL)/Test	Ref.
*Murraya koenigii*	mahanimbine	2.5 µg/mL/-	0.14 µg/mL/-	H	L/hemagglutination	[64]
	girinimbine	1.1 µg/mL/-		H		
	koenimbine	3.8 µg/mL/-		H		
	mahanimbinol	4 µg/mL/-		H		
	girinimbinol	1.2 µg/mL/-		H		
	mahanine	3.5 µg/mL/-		H		
	murrayanine	4.6 µg/mL/-		H		
	chemical derivates of the above compounds	0.6–3.9 µg/mL/-		H		
*Murraya koenigii*	bergapten	4 µg/mL/-	1.9/-	H	L/hemagglutination	[52]
	imperatorin	6 µg/mL/-		H		
	isoimperatorin	3.1 µg/mL/-		H		
	5-methoxyimperatorin	15.2 µg/mL/-		H		
	heraclenin	11 µg/mL/-		H		
	byakangelicol	22 µg/mL/-		H		
	isogosferol	2 µg/mL/-		H		
	8-geranylosypsoralen	22 µg/mL/-		H		
	indicolatone	2.1 µg/mL/-		H		
	β-sitosterol	5.8 µg/mL/-		H		
*Clausena lansium*	imperatorin *	6/-	1.9/-	H	N.t. (only for diabetes)	[50]
	phellopterin *	15.2/-		H		
	3-phormylcarbazole *	3.6/-		H		
	Chalepin *	22.4/-		H		
*Peganum harmala*	harmine	-/30	-/50	H	N.t.	[57]
	harmaline	-/100		M		
Commercial compounds present in several Lamiaceae plants	1,8-cineole, linalool, carvacrol, α-terpineol, camphor, borneol	>100/-	1	L	L (>100)	[60]
	thymol	38.4/-		H	L (>100)	
	linalyl acetate	32.2/-		H	L (>100)	
	γ-terpinene	43.8/-		H	L (>100)	
	p-cymene	59.7/-		M	L (>100)	
	caryophyllene oxide	100/-		M	L (>100)	
	α-pinene	44.2/-		H	M (88.9)	
	β-pinene	29.6/-		H	M (88.2)	
	D-fenchone	61.9/-		M	L (>100)	
	β-caryophyllene	86.1/-		M	M (60.8)	
	α and β-thujone	17.3/-		H	L (>100)	
	4-terpineol	41.5/-		H	L (>100)	
	camphene	24/-		H	M (74.8)	
Only compound	carvacrol	0.39/-	2.17/-	H	L (toxicity at doses higher than IC_50_ for *T. gallinae*)	[65]
	carvacrol (nanoemulsion)	0.27/-		H	L	

* Compounds tested in vivo for other purposes (not experimental infections with *T. gallinae*). MTZ: Metronidazole as a reference compound. IC_50_: Inhibitory Concentration 50. MIC: Minimum Inhibitory Concentration. N.t.: not tested. Ref: reference. H: high, MIC or: IC_50_ ≤ 50 µg/mL (High), M: moderate, MIC or IC_50_ > 50 and ≤ 100 µg/mL, L: low, MIC or IC_50_ > 100 µg/mL. For cytotoxicity L: low, IC_50_ > 100 µg/mL, M: moderate, IC_50_ > 50 and ≤100 µg/mL, or H: high, IC_50_ ≤ 50 µg/mL.

## Data Availability

Not applicable.

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
