# Peer review of "Avian Oropharyngeal Trichomonosis: Treatment, Failures and Alternatives, a Systematic Review"

_microorganisms, 2022, doi:10.3390/microorganisms10112297_

Round 1

Reviewer 1 Report

The manuscript microorganisms-2006071 entitled "Title: Avian oropharyngeal trichomonosis: treatment, failures and alternatives, a systematic review." is a well written systematic review about treatment options in case of trichomonosis in birds. Data acquisition and selection of data for the systemic review are presented clearly and are comprehensible. The structure of the comparative presentation, starting with the established therapeutics, through the development of resistance to the nitroimidazoles and finally an overview and outlook of alternatives, is very reader-friendly.

There are only minor editing requests.

Please italicize all taxonomic names throughout the manuscript.

Please put a space between number and unit throughout the manuscript (see for instance line 159 or line 164).

Line 227: Please write a point between the numbers instead of a comma.

Table 3/4: What does the numbers (and letter in table 4) mean after the listed reference?

Line 304: change kg in lower case 

Author Response

We would like to thank the reviewer to take the time to revise our manuscript and for constructive comments.

All the taxonomic names have been italicized throughout the manuscript. We apologize for these mistakes, although in the version we uploaded all the names were in italics, there must be some “mysterious” issues with the platform to upload the manuscript. Anyhow, thanks for pointing out this issue.

A space was added between the number and the unit (mg) at the mentioned places, and also reviewed along the manuscript.

A point was used to replace the comma at line 227.

We are sorry for the mistakes in table 3 & 4. We suppose that they are mistakes not fully corrected from previous versions. They have been eliminated now.

Reviewer 2 Report

Authors have provided an extensive and systematic review. 

Minor points may be only requested to improve the presentation of the study.

Conclusions may also provide some data on the effectiveness of the alternative feed additives, such as plant extracts or essential oils.

line 450, the word more is not necessary and may be deleted.

Studies on the oregano essential oil may be searched furtherly to add to the discussion part.

Author Response

We would like to thank the reviewer for the constructive comments and also for taking the time to review our manuscript. We have followed your recommendations.

-The names of the most active plants (extract and compounds) were included in the conclusions.

- The word “more” was eliminated from line 450.

- As suggested by the reviewer, we have extended the search to the specific word “oregano” and “trichomonas gallinae”, but we only found the article “EFFECTS OF OREGANO (ORIGANUM VULGARE) AS A DIETARY SUPPLEMENT IN TURKEY FARMING-REVIEW. N. N. N BOZAKOVA. JOURNAL OF HYGIENIC ENGINEERING AND DESIGN 38:219-224 2022”. However, although we found this article in our initial search and found it really interesting, it is a review article, and we could not include it in the present manuscript because reviews were a cause of exclusion in our search strategy. We agree with the reviewer that Origanum is one of the best essential oils, since we already observed its efficacy in a previous study.